# Correlation between Chemical Profile of Georgian Propolis Extracts and Their Activity against *Helicobacter pylori*

**DOI:** 10.3390/molecules28031374

**Published:** 2023-02-01

**Authors:** Jarosław Widelski, Piotr Okińczyc, Katarzyna Suśniak, Anna Malm, Anna Bozhadze, Malkhaz Jokhadze, Izabela Korona-Głowniak

**Affiliations:** 1Department of Pharmacognosy with Medicinal Plants Garden, Lublin Medical University, ul. Chodźki 1, 20-093 Lublin, Poland; 2Department of Pharmacognosy and Herbal Medicines, Wrocław Medical University, ul. Borowska 211a, 50-556 Wrocław, Poland; 3Department of Pharmaceutical Microbiology, Medical University of Lublin, 20-093 Lublin, Poland; 4Department of Pharmacognosy, Tbilisi State Medical University, 33 Vazha-Pshavela Ave, 0186 Tbilisi, Georgia; 5Department of Pharmaceutical Botany, Tbilisi State Medical University, 33 Vazha-Pshavela Ave, 0186 Tbilisi, Georgia

**Keywords:** propolis, Georgia, hydroethanolic extracts *Helicobacter pylori*, antibacterial, PCA, dendrogram, black poplar, flavonoids

## Abstract

*Helicobacter pylori* (*H. pylori*) is considered the most common bacterial pathogen colonizing stomach mucosa of almost half the world’s population and is associated with various gastrointestinal diseases (from digestive problems and ulcers to gastric cancer). A lack of new drugs and a growing number of *H. pylori* antibiotic-resistant strains is a serious therapeutic problem.As a mixture of natural compounds, propolis has antimicrobial activity based on high concentrations of bioactive polyphenols (mainly flavonoids and phenolic acid derivates). The chemical composition of tested Georgian propolis is characterized by the presence of flavonoids aglycones, and phenolic acid monoesters, e.g., pinobanksin-5-methyl ether, pinobanksin, chrysin, pinocembrin, galangin, pinobanksin-3-*O*-acetate, pinostrobin and pinobanksin-3-*O*-butanoate, or isobutanoate and methoxycinnamic acid cinnamyl ester. The anti-*H. pylori* activity of 70% ethanol water extracts of 10 Georgian propolis samples was evaluated in vitro by MIC (minimal inhibitory concentration) against the reference strain (*H. pylori* ATCC 43504) and 10 clinical strains with different antibiotic-resistance patterns. The strongest anti-*Helicobacter* activity (MIC and MBC = 31.3 µg/mL) was observed for propolis from Orgora, Ota, and Vardzia and two from Khaheti. Lower levels of activity (MIC = 62.5 µg/mL) were found in propolis obtained from Qvakhreli and Pasanauri, while the lowest effect was observed for Norio and Mestia (MIC = 125.0 µg/mL). However, despite differences in MIC, all evaluated samples exhibited bactericidal activity. We selected the most active propolis samples for assessment of urease inhibition property. Enzyme activity was inhibited by propolis extracts, with IC_50_ ranging from 4.01 to 1484.8 µg/mL. Principal component analysis (PCA) and hierarchical fuzzy clustering (dendrograms) coupled with matrix correlation analysis exhibited that the strongest anti-*Helicobacter* activity was connected with black poplar origin and high flavonoid content of propolis. Samples with lower activity contained higher presence of aspen markers and/or dominance of non-flavonoid polyphenols over flavonoids. In summary, Georgian propolis can be regarded as a source bioactive compounds that can be used as adjuvant in therapy of *H. pylori* infection.

## 1. Introduction

Identified in 1982, *Helicobacter pylori* (*H. pylori*) is a pathogen that has acquired resistance to different drugs used in conventional therapy, which is of worldwide concern [1]. These data are reflected in the World Health Organisation report indicating that *H. pylori* is priority 2 on its global list of antibiotic-resistant bacteria, based on its increased resistance to clarithromycin [2].

Infection caused by *H. pylori* is prevalent in almost 50% of the world’s population. Thus, it is a very important public health problem in most countries, especially well-developed ones. Since this bacterial infection is correlated with a wide spectrum of gastric disturbances, starting from mild discomfort, such as superficial gastritis, to serious illnesses, including chronic atrophic gastritis, peptic (gastric or duodenal) ulcers, and even gastric cancer, there is great interest in understanding possible mechanisms of prevention [3]. The severity and progression of gastric cancer depend on the presence of specific *H. pylori* virulence factors. *H. pylori* infection influences cellular components that are associated with epithelial–mesenchymal transition progression [4].

Usually, *H. pylori* is contracted in childhood, and oral–oral or oral–fecal routes are the most likely ways of transmission of bacteria between individuals. Children become healthy carriers of the pathogen and may not have any symptoms of infection, or it can occur during adulthood [5,6,7].

The stomach is an excessively hostile environment, predominantly to the presence of gastric mucosa, as well as low pH due to high concentration of hydrochloric acid [8]. However, *H. pylori* possess several virulence factors that are necessary to colonize this extreme environment and to survive in it. Currently, among vacuolizing cytotoxin and pathogenicity gene products (cagPAI—cag pathogenicity island), urease is considered one of the major virulence factors of this pathogen [1,9]. The urease enzyme is capable of hydrolyzing urea present under the physiological state in the acidic medium of the stomach. Ammonia produced during the reaction acts as a receptor for H^+^ cations that in consequence generate neutral (higher) pH in the intracellular environment [1,3,10].

Additionally, urease activity is crucial for nutrient acquisition and is generally essential for the capacity of *H. pylori* to colonize the gastric epithelium [11,12]. Urease and the product of its reaction, ammonia, are responsible for the destabilization of the protective mucus layer, what in consequence leads to lesions on lining cells. Urease also causes a strong immune response, is chemotactic to phagocytes [13], and increases the production of cytokines, among them interleukins—IL-1β, IL-6, and IL-8—and TNF-α (tumoral necrosis factor-alpha) [3,14]. All of these activities of urease are grounds for local inflammatory lesions [10].

It is worth mentioning that commercially accessible inhibitors of urease, such as imidazoles, hydroxamic acid derivatives, or a group of phosphorodiamidates, are characterized by high toxicity and poor stability, which restrict their clinical use [12,15].

These facts, together with the failure of *H. pylori* eradication in many countries prompted by increasing antibiotic resistance, are good reasons for the search for safe and effective nonantibiotic therapeutic agents for treating this kind of infection.

Natural products seem to be a good alternative source of active substances and show antimicrobial activity against *H. pylori*, as well as inhibiting urease activity and preventing harmful consequences.

Components of natural origin, including plant extracts or their constituents and bee products, exert anti-*H. pylori* activity, gastroprotective action, and an anti-inflammatory effect [16,17]. For example, in studies conducted by Korona-Glowniak et al. [18], in vitro activity of essential oils (obtained from thyme, lemongrass, cedarwood, basil, and other essential oil–bearing plants) against *H. pylori* growth as well as urease activity were proved.

One of the natural products known as a natural remedy for various infectious diseases, among them *H. pylori* infection, is propolis [19,20,21]. Propolis, known also as “bee glue” is a resinous mixture that different species of honeybees produce from plant exudates gathered from tree buds and flowers, as well as other botanical sources, and bee salivary secretions, wax, and pollen [22]. The chemical composition of propolis is heterogeneous and may vary according to the geographical location and thus plant precursors as well as the time of collection [8]. Among the main components of propolis resin (50%), wax (up to 30%), essential oils (10%), and organic compounds can be listed. More than 300 organic compounds have been identified in propolis obtained from different geographical locations, mainly flavonoids, phenolic compounds, esters and terpenes [23]. However, the presence of such polyphenols as caffeic acid esters, chrysin, galangin, pinocembrin, pinostrombin and quercetin in propolis extracts has been shown to be essential for biological activities [24].

Except for broad antimicrobial activity [25,26] propolis extracts have effects on all pathophysiological symptoms of *H. pylori* infection. The results of recent studies indicate that the use of Korean propolis, which has anti-inflammatory and antioxidative effects, is promising for the prevention of *H. pylori*-induced gastric damage [17]. Studies on *Helicobacter pylori*-infected gastric mucosal injury mice model demonstrate the beneficial effects of Korean propolis on inflammation through the inhibition of NF-κB signaling and inhibition of *H. pylori* growth [27]. These results showed that propolis seems not only to be helpful in treating *H.pylori* infection but also in alleviating adverse reactions during and after long antibiotic therapy.

The main goals of this study were to evaluate for the first time the anti-*H. pylori* activity of tested propolis samples obtained from different locations of Georgia, assess the relationship between the polyphenolic profile of propolis, plant origin, and antimicrobial activity against *H. pylori*, and evaluate the inhibition activity of selected propolis samples toward urease enzyme.

## 2. Results and Discussion

### 2.1. Antibacterial Activity of Tested Georgian Propolis Extracts against H. pylori

Despite scientific reports on propolis antimicrobial properties being quite extensive, to the best of knowledge, there have been no studies on the anti-*Helicobacter* activity of Georgian propolis from different locations of Georgia. Moreover, there are few articles about the activity and chemistry of propolis from Georgia. Our group assessed 10 propolis hydroethanolic extracts (70 EEP; 70:30, ethanol:water, *v*/*v*,) obtained from different propolis samples derived from various parts of Georgia. The results of antibacterial assays are presented in Table 1. The highest activity against the reference *H. pylori* strain was expressed by PE from Ota, Ogora, and Vardzia and two unknown samples (in general from the Kakheti region), with MIC of 31.3 µg/mL. It is worth mentioning that the rest of the tested 70 EEP possessed good bioactivity, as their MICs were in the range of 26–125 µg/mL based on bioactivity criteria established by O’Donnell et al. [28]. The 70 EEP obtained from samples collected in Pasanauri, Qvakhreli and Aspindza showed weaker activity (MIC = 62.5 µg/mL), though better than the other 70 EEP from Norio and Mestia, (MIC = 125 µg/mL). Surprisingly, for all 70 EEP evaluated against *H. pylori*, the MBC/MIC ratio was 1, which confirmed the bactericidal activity of tested PEs (Table 1). In relation to the reports of other authors, propolis extracts exhibit bacteriostatic rather than bactericidal effects against sensitive bacterial strains [29].

The antimicrobial activity of the 10 70 EEP was evaluated against 10 clinical strains for sensitivity to antibiotics (5 strains) and 5 resistant strains to at least 1 antibiotic (metronidazole, rifampicin, clarithromycin, tetracycline or levofloxacin). Propolis extract activity was defined on the basis of MIC_50_ and MIC_90_, i.e., MIC values inhibiting 50% or 90% of the studied clinical strains, respectively. It was shown that the antibacterial activities of the tested propolises were similar, irrespective of antibiotic susceptibility of clinical *H. pylori* strains, suggesting different mechanisms of action (Table 1).

There are several papers related to antimicrobial activity against different reference strains and clinical isolates of *H. pylori* exerted by propolis extracts gained from different geographical locations, as well as their fractions and isolated compounds. The difficulty in comparing the results of studies on activity against *H. pylori* and other pathogens in general is differences in methodology. The present study used the broth dilution assay to determine MIC, while other authors relied on the agar-well diffusion assay or disk diffusion, which are rather qualitative methods and less suited for testing such a fastidious bacterial species as *H. pylori* [12,30].

In our previous research, we determined the MIC/MBC of propolis extract samples from Poland, Ukraine (4 samples each), Kazakhstan, and Greece (1 sample each) against *H. pylori* (ATCC 43504) [31]. The obtained results were in the range of 20.0 to 62.5 µg/mL, and the highest activity against the reference *H. pylori* strain was shown by 70 EPP from Ukraine (UK3), with MIC of 20 µg/mL, and one PE from Poland (PLS2) had the lowest activity—62.5 µg/mL. The mentioned activity of 70 EEP is similar to the current research of our group, as well as for the studies conducted by Santiago et al. [8] for Brazilian red propolis. Hydroalcoholic extract of red propolis from Brazil showed activity against *H. pylori* (ATCC 43526), with MIC of 50 µg/mL, and a clinical isolate of *H. pylori* with MIC of 100 µg/mL. The MBC values were the same [8]. Nineteen PEs (ethanolic and propylene glycol) obtained from a different location in northern Spain (Basque Country) were tested by Bonvehí et al. [32] and exhibited very weak activity against *H. pylori* (MIC from 6 to 14 mg/mL). In another study, 70% ethanolic PE from a sample produced by an Indonesian stingless bee (*Trigona* spp.) were tested on 10 clinical strains of *H. pylori* isolated from dyspeptic patients [33]. The tested strains were clarithromycin- and metronidazole-resistant. There was very weak activity of the tested extracts, in the range of 1024–8192 µg/mL. Notably, in that research, the Indonesian extracts showed a very interesting result of an additive effect against *H. pylori* used together with clarithromycin or metronidazole. According to the authors’ suggestion, the fact that propolis has the ability of urease inhibition [12] and HpPDF (*Helicobacter pylori* peptide deformylase) inhibition [34] might be the result of an additive effect with an anti-*H. pylori* regimen. Moreover, the additive effect was still observed in the resistant strains [33]. An optimistic conclusion of this study is that even extracts with weak activity (high MIC value) can be used in therapy for *H. pylori* infection as an adjuvant. This will reduce antibiotic doses in the eradication of the pathogen, which will reduce the cost of therapy and its safety (lower toxicity and side effects) [33].

The activity of a 30% ethanolic extract from Bulgarian propolis against 94 *H. pylori* strains isolated from patients with gastroduodenal diseases was assessed by Boyanova et al. [30]. For evaluation activity in vitro, three methods were used: agar-well diffusion, agar dilution, and disk diffusion. Extracts (30 µL containing 9 mg of propolis) inhibited the growth of 82.8% of *H. pylori* strains, while for 90 µL extracts (containing 27 mg of propolis), the inhibition rate was 100%. The results of the study showed strong, dose-dependent anti-*H. pylori* activity of propolis from Bulgaria [30].

The in vitro antimicrobial activity of forty-two honey and eight ethanolic extracts (obtained from six Turkish propolis and red Brazilian propolis) was investigated against 16 microorganisms (including *H. pylori*) by Kolayli and coworkers [35]. For the determination of antimicrobial activity, the authors used the agar well diffusion method. All tested extracts showed significant inhibition against all studied microorganisms, but the widest inhibition zone (the strongest activity) was found against *H. pylori*, which is a fastidious pathogen. The authors underlined that extracts obtained from poplar-type propolis samples (Turkish propolis) have much better inhibition effects than numerous honey samples [35]. These conclusions are in line with those we have drawn from the results presented in this paper, as well as a previous one [31].

A majority of papers concerning the various biological or pharmacological activity of plant extracts, but especially bee product extracts (including propolis), are aimed at the parameter describing the particular activity. In the case of antibacterial activity, it is MIC or MBC values presenting quantitative methods of testing. When propolis activity is studied, it is common to look for correlations between activity (antioxidant, antimicrobial, or inhibition of important enzymes) and total phenolic content or total flavonoid content.

Our publication is an attempt to combine the results of studies on the inhibition of *H. pylori* growth by propolis extracts and qualitative analysis of their composition by using chromatographic and spectral analysis (LC-MS). An exceptional work is research performed by Romero et al. [20] concerning propolis polyphenols’ effect on the viability and structure of *H. pylori*. From the propolis sample from Chile, using CPC and preparative HPLC, the authors isolated four major polyphenols: chrysin, pinocembrin, galangin and caffeic acid phenylethyl ester (CAPE). These compounds inhibited both reference and clinical *H, pylori* strains (MIC 256–1024 µg/mL), with CAPE being the most active. Fractional inhibitory concentration was 64–512 µg/mL, and chrysin with galangin exerted synergistic effects. Surprisingly, the main compounds isolated from the propolis sample of Chile showed moderate to mild antibacterial activity upon tested *H. pylori* strains. Nevertheless, the synergistic effect of two polyphenols, chrysin and galangin, indicates that the synergy of all polyphenols, flavonoids, phenolic acids, and their esters is responsible for the antimicrobial activity of propolis [20]. The chemical composition of the investigated Georgian propolis with the same polyphenolic compounds can explain the good activity against *H. pylori.* However, results obtained by Romero and coauthors demonstrate that in such a complex mixture as propolis, not only must the presence of canonical antibacterial compounds be determined but also the ratio and potential interaction among them [20].

Romero et al. [20] confirmed, by morphological study through transmission electron microscopy, a possible mechanism of the inhibition effect of polyphenols on bacteria growth. The mechanism of their action is based on lysis and vesicle formation, which is similar to the activity of amoxicillin, namely, an inhibitory effect on bacterial peptidoglycan synthesis [20]. Similar findings were reported by Eumkeb et al. [36], who demonstrated that galangin and other flavonoids produced outer membrane detachment on *E. coli*, and it is likely such an effect is due to internal damage of the peptidoglycan layer.

Krzyżek et al. [37] observed that sub–minimal inhibitory concentrations (sub-MICs) of this polyphenolic natural compound myricetin, have the ability to slow the process of transformation into coccoid forms and reduce biofilm formation of *H. pylori*. The ability of the pathogen to change from a spiral to a coccoid form is one of the most important factors related to its eradication failures. In addition, the authors noted that exposure of *H. pylori* to the sub-MIC of myricetin enabled 4- to 16-fold reduction in the MIC of all classically used antibiotics (amoxicillin, clarithromycin, tetracycline, metronidazole and levofloxacin), underscoring the importance of compounds and natural products as adjuncts in the treatment of gastric diseases caused by *H. pylori* infection [37].

According to RT-qPCR studies carried out by the authors, myricetin decreased expression of the genes *csd3*, *csd6*, *csd4* and *amiA*, which are involved in muropeptide monomer shortening and play important role in spiral-to-coccoid transition [37].

Reported data indicate that propolis extracts containing numerous polyphenolic compounds with multifactorial activity can be a natural alternative to present eradication methods, which are becoming less effective due to bacterial resistance and cost, as well as side effects of classical therapies.

### 2.2. Urease Inhibitory Activity of Tested Georgian Propolis Extracts

The results for the assessment of the tested 70 EEP from Georgian propolis are listed in Table 2.

Our study is the first to investigate the effects of 70 EEP obtained from propolis samples collected in Georgia, a natural bee product used in the treatment of gastric diseases, on *H. pylori* growth in vitro, as well as its enzyme urease activity, which is crucial for its ability to colonize the stomach. Urease activity dropped to 42.5% and 78.3% of the control in concentration of 1 mg/mL of tested propolises. Results of bioassays showed IC_50_ values for 70 EEP ranging from 4.01 to 3859.23 µg/mL and IC_50_ of 92.7 µg/mL for thiourea (reference inhibitor).

Results obtained in our experiments are very promising in comparison to other studies of 70 EEP inhibitory activity and indicate that searching for novel, natural urease inhibitors among bee products is the proper direction. Baltsas et al. [12] tested 15 PEs obtained from beekeepers from a different region of Turkey for inhibition of urease activity. The inhibition concentration (IC_50_) was in the range of 0.260 to 1.525 mg per mL, similar to the results presented in this study. Inhibition activity of 70 EEP from Georgia was decidedly weaker than other natural products, e.g., essential oils. For example, Oregano essential oil (MIC = 31.3 µg/mL) showed IC_50_ against *H. pylori* urease of 208.3 µg/mL [18]. Moreover, the most active essential oil obtained from cedarwood had IC_50_ of 5.3 µg/mL (MIC = 15.6 µg/mL).

In studies conducted by Can [38], 11 propolis samples from the Marmara region of Turkey were investigated. The antiurease activity of IC_50_ ranged from 1.110 to 5.870 mg/mL, and the authors suggested that this indicated good inhibitory activity of the tested propolis extracts [38]. In fact, the activities of tested propolis by Can were similar to that presented in the previous paper [39], where enzyme inhibition of urease was examined by different bee products: honey, pollen and propolis. The urease inhibition values (IC_50_) changed from 7.02 to 33.25 mg/mL, 5.00 to 8.78, and 0.16 to 1.98 mg/mL in the honey, pollen and propolis samples, respectively [39].

*H. pylori* urease is a crucial enzyme for this bacterium surviving in an acidic environment of the stomach. The main defence against infection of this pathogen is prevention from the adhesion of bacteria to gastric mucosa, which can be gained by urease inhibition [12]. Georgian propolis, especially in comparison with other published data, seems to be a good agent with the ability to inhibit urease among all bee products.

### 2.3. Impact of Components of Propolis Extracts on Anti-Helicobacter Activity

#### 2.3.1. Composition of Georgian Propolises

Composition of propolis extracts was analyzed in detail in our previous research [40]. In this study, the concentration of components was calculated as percentage of the whole UV chromatogram area (280 nm). The range of the UV chromatogram was chosen due to the spectral parameters of the main polyphenols, all of which exhibited strong absorbance in this range. These results were also used for further statistical calculation to correlate the phytochemical composition of tested propolises with anti-*H.pylori* activity. Generally, the strongest peaks belonged to flavonoid aglycones, as well as some phenolic acid monoesters (Table 3).

#### 2.3.2. Correlation Matrices

Results of matrix correlations are presented in Appendix A. Total phenolic content (TP) and flavonoid content (TF) were described in previous research [40]. Generally, the activity against *H. pylori* was negatively correlated with TF and TP of extracts. as well as some single components (pinobanksin-5-methyl ether, pinobanksin, chrysin, pinocembrin, galangin, pinobanksin-3-*O*-acetate, pinostrobin and pinobanksin-3-*O*-butanoate or isobutanoate and methoxycinnamic acid cinnamyl ester) [Figure 1]. Some phenolic glycerides and vanillin, *p*-coumaric, cinnamic, and ferulic acids exhibited a positive correlation with activity against *H. pylori*. These results may suggest that flavonoid aglycones exhibit the most important role in the anti-*Helicobacter* activity of propolis extracts. Apart from these, the presence of non-flavonoid polyphenols, especially phenolic acid glycerides, suggested weaker activity of whole extracts.

#### 2.3.3. Principal Component Analysis and Hierarchical Fuzzy Clustering

Results of PCA are presented in Figure 2 and Figure 3. The dendrogram of hierarchical fuzzy clustering analysis is presented in Figure 4.

The model based on two factors was sufficient to show 80.9% of presentation quality. Moreover, it is important to note that the most active samples against *H. pylori* were represented in one small cluster (MIC = 31.30 µg/mL). All of these samples exhibited black poplar origin with a dominance of flavonoids over the rest of the polyphenols. Samples outside this cluster exhibited weaker anti-*Helicobacter* activity. In summary, this result suggests that a stronger presence of aspen markers (Mestia, Norio, Passanauri, Aspindza) and the dominance of non-flavonoid polyphenols over flavonoid in poplar-origin propolis (Qvakhreli) is connected with lower anti-*H.pylori* activity. Similar results were presented in previous research for Gram-positive bacteria [31,40]. Moreover, when hierarchical fuzzy clustering was used, almost the same results as PCA were obtained (Figure 4).

## 3. Materials and Methods

### 3.1. Propolis and Reagents

Propolis from the following regions of Georgia was obtained in 2020: Aspindza, Norio, Pasanauri, Mestia, Orgora, Vardzia, Ota, Qvakhreli, and two unknown locations (in Kaheti). The propolis was frozen in liquid nitrogen and crushed in mortar. Procedures were repeated three times. Before extraction, ground propolis was stored in sealed containers under −20 °C.

LiChrosolv^®^ hypergrade eluents for LC-MS (acetonitrile, water, methanol), DPPH (2,2-diphenyl-1-picrylhydrazyl), TPTZ (complex of 2,4,6-Tri(2-pyridyl)-s-triazine), iron(II) sulfate heptahydrate, and aluminium chloride hexahydrate were purchased from Merck (Darmstad, Germany). Folin–Ciocâlteu reagent and ethanol (analytical grade) were purchased from ChemPur (Piekary Śląskie, Poland). Disodium hydrogen phosphate and sodium chloride were obtained obtained from POCH (Gliwice, Poland). Mueller–Hinton agar and Sabouraud agar were obtained from Oxoid (Hampshire, UK).

### 3.2. Preparation of Propolis Extracts (70 EEP)

The ground research material was extracted by ethanol in water (70:30; *v*/*v*) in a proportion of 1:10 (1.0 g of propolis per 10 mL of solution). Extraction was performed in an ultrasonic bath (Sonorex, Bandelin, Germany). Extraction conditions were set at 20 °C for 45 min and 756 W (90% of ultrasound bath power). Next, extracts were stored at room temperature for 12 h and then filtered through Whatman no. 10 paper (Cytiva, Marlborough, MA, USA). For all samples, extraction efficiency was calculated as percentage of dry extract mass in crude propolis.

### 3.3. UHPLC-DAD-MS/MS Profile of Propolis Extracts

UHPLC-DAD-MS/MS analysis was performed according to previously developed method [40]. uHPLC-DAD-MS/MS was carried out using a compact QqTOF MS/MS detector (Bruker, Darmstadt, Germany). The MS detector was used in electrospray negative mode. Conditions of analysis were: ion source temperature 210 °C, nebulizer gas pressure 2.0 bar, and dry gas (nitrogen) flow 8.01 L/min. The capillary voltage was programmed at 4.5 kV. The collision energy was set at 8.0 eV. Internal calibration was obtained with 10 mM solution of sodium formate. For ESI-MS/MS experiments, collision energy was set at 35.0 eV and nitrogen was used as collision gas. The scan range was set as 30–1300 m/Z.

### 3.4. Colorimetric Assays of Propolis Extracts

Colorimetric assays was performed using extracts described in Section 3.3. Before proper measurements, preliminary analysis with different dilutions of basic extracts two to ten times was carried out to obtain the most appropriate concentration for every assay.

Antiradical activity (DPPH test) and total antioxidant activity (FRAP assay) and total phenolic content (TP) and total flavonoid content (FC) assays were performed according to previously described methods [40].

### 3.5. Antibacterial Activity Assay

The 70 EEP dissolved in dimethylosulfoxide (DMSO) were screened for antibacterial activity by microdilution broth method according to the European Committee on Antimicrobial Susceptibility Testing (EUCAST) (www.eucast.org (accessed on 1 December 2022)) using Mueller–Hinton broth with 7% lysed horse blood, as described elsewhere [18].

*H. pylori* ATCC 43504 was obtained from American Type Culture Collection (Rockville, MD, USA). The 10 clinical *H. pylori* strains used in the study were isolated from patients with gastrointestinal disorders, tested and described elsewhere [41]. Five strains were susceptible to all tested antibiotics and of 5 strains, 2 were resistant to 1 antibiotic (metronidazole), 1 resistant to 2 antibiotics (metronidazole + rifampicin), 1 resistant to clarithromycin + levofloxacin + metronidazole, and 1 resistant to clarithromycin + levofloxacin + metronidazole + rifampicin.

The MIC for *H. pylori* strains was determined using twofold microdilution at extract concentration ranging from 1000 to 1.95 mg/L with bacterial inocula of 3 McFarland standard. After incubation at 35 °C for 72 h under microaerophilic conditions (5% O_2_, 15% CO_2_ and 80% N_2_), the growth of *H. pylori* was visualized with the addition of 10 µL 0.04% resazurin. The MIC endpoint was recorded after 4 h incubation as the lowest concentration of extract that completely inhibited growth [18]. Appropriate DMSO control, a positive control (inoculum without the tested 70 EEP), and a negative control (the tested 70 EEP without inoculum) were included with each microplate. MIC 50 and MIC 90 were defined as the minimum concentration at which 50% and 90% of the isolates were inhibited, respectively.

Minimal bactericidal concentration (MBC) was obtained by culture of 5 µL from each well that showed growth inhibition, from the last positive one, and from the growth control onto recommended agar plates. The plates were incubated at 35 °C for 72 h under microaerophilic conditions. The experiments were repeated in triplicate. Representative data are presented.

### 3.6. Urease Inhibitory Assay

Briefly, *H. pylori* samples were incubated for 72 h in the MH broth with 7% horse serum (Sigma-Merk, St. Louis, MO, USA) in microaerophilic conditions. Bacterial biomass was collected by centrifugation at 4000× *g* in 4 °C for 10 min, then the cells were dissolved in ice-cold phosphate buffer (pH 7.3) with a protease inhibitor cocktail (Sigma). Urease enzyme was prepared by disturbing *H. pylori* cells by sonication, followed by centrifugation at 12,000× *g* and 4 °C for 10 min.

Initial urease inhibitory activity of all the obtained extracts was evaluated at a concentration of 2 mg/mL with the modified Berthelot spectrophotometric method, with phenol–hypochlorite reaction at absorbance of 570 nm. The enzyme reaction was activated in 96-well plates by mixing the appropriate volume of 2% urea, sodium phosphate buffer solution (100 μL), and different concentrations (2000–3.9 μg/mL) of propolis extract, and the reaction mixture was incubated for 15 min at 37 °C. Then, ammonia concentration was determined using the Berthelot method. The amount of ammonia was equivalent to urea hydrolysis using the urease enzyme. The experiments were performed in triplicate. Activity of uninhibited urease was chosen as the control activity of 100% [42]. Inhibition rate was calculated following the formula I % = (1-average with inhibitors/average activity without inhibitors) × 100%. IC_50_ is expressed as the concentration of inhibitor that decreased urease activity by 50% and calculated by plotting the percentage of inhibition using the internet IC_50_ Calculator (AAT Bioquest, Pleasanton, CA, USA).

### 3.7. Statistical Analysis

Statistical analyses were performed using Statistica 14.0.0.5 software (Tibco Sofware Inc., Palo Alto, CA, USA). Correlations between composition and activity were analysed by searching for correlations in the prepared matrix. This was composed of percentage of UV chromatograms (280 nm) relative peak area of 70 EEP, antimicrobial activity (MIC against *Helicobacter pylori*), and colorimetric test values (DPPH, FRAP, TP and TF). Substances of at least 1% of relative area (in any sample) were used to construct the matrix, attached in Appendix A.

Statistical analyses included evaluation of impact of composition on antimicrobial activity (Pearson correlation and R^2^ parameters) as well as principal components and hierarchical fuzzy clustering analyses. PCA and dendrogram matrices were composed of substances only without MIC values. These analyses were used for classification of propolis plant origin.

## 4. Conclusions

The activity of 70 EEP from Georgian propolis samples against *H. pylori* is presented in this paper for the first time. Tested extracts exhibited good activity against this pathogen (MIC/MBC from 31.3 to 125 µg/mL). The current study has confirmed that Georgian propolises possess antiurease activity (1 mg/mL of propolis extracts inhibited urease activity in the range of 42.5% to 78% of control activity) indicating a presumptive mechanism of anti-*H. pylori* activity of propolises. Flavonoid aglycones exhibited the most important role in anti-*Helicobacter* activity of propolis extracts. The isolation of active components from tested propolis samples should be attempted. Nevertheless, the tested propolis samples can be regarded as an efficient component (as adjuvant) during *H. pylori* infection therapy.

## Figures and Tables

**Figure 1 molecules-28-01374-f001:**
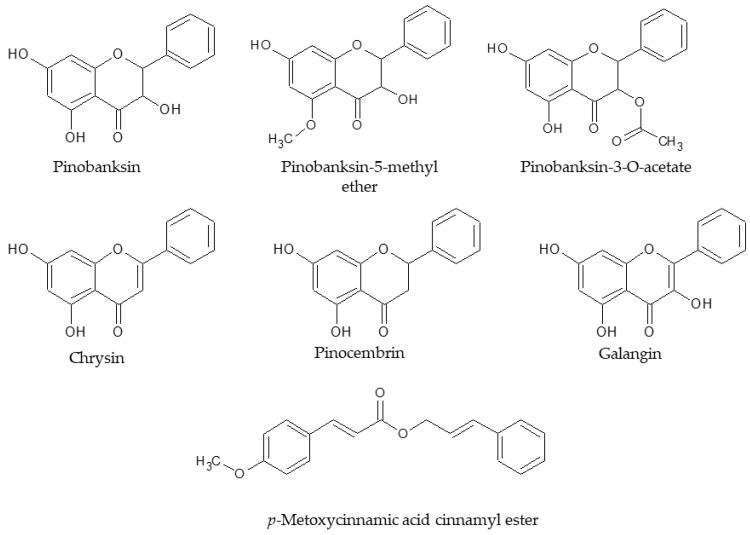
Structure of components connected with the anti-*Helicobacter* activity of propolis extracts.

**Figure 2 molecules-28-01374-f002:**
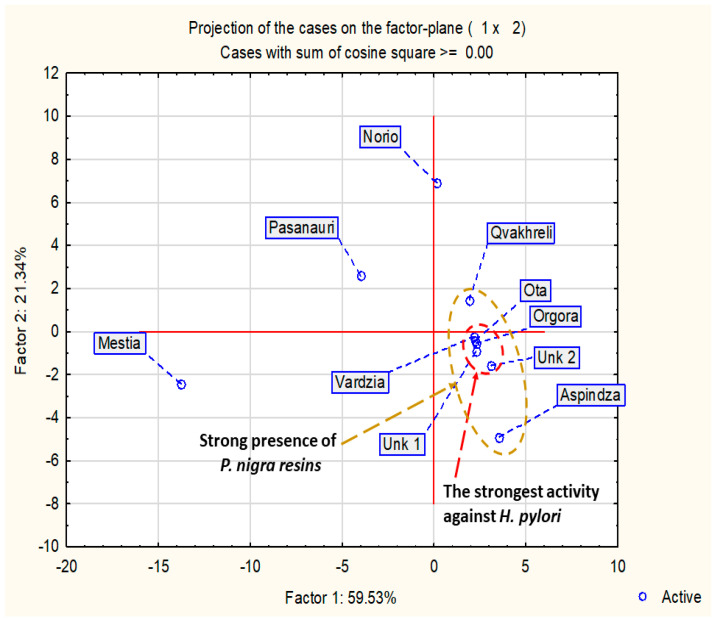
Projection of cases on the factor plane.

**Figure 3 molecules-28-01374-f003:**
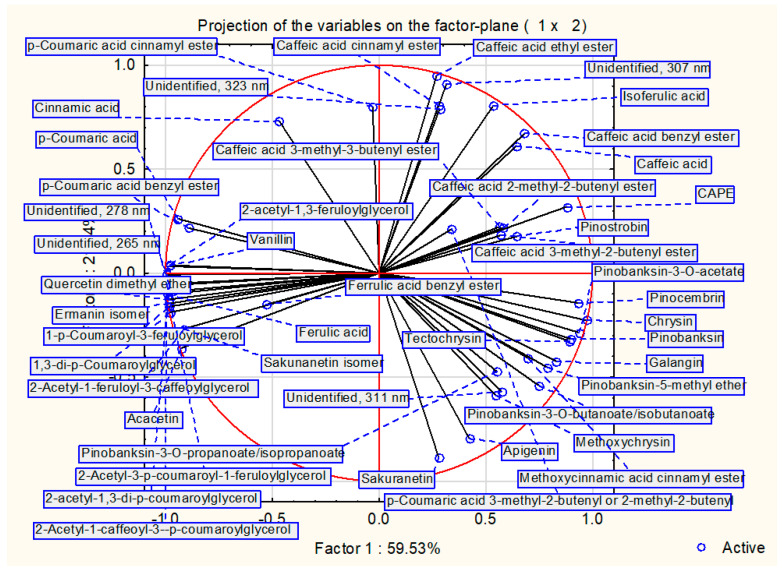
Projection of variables on the factor plane.

**Figure 4 molecules-28-01374-f004:**
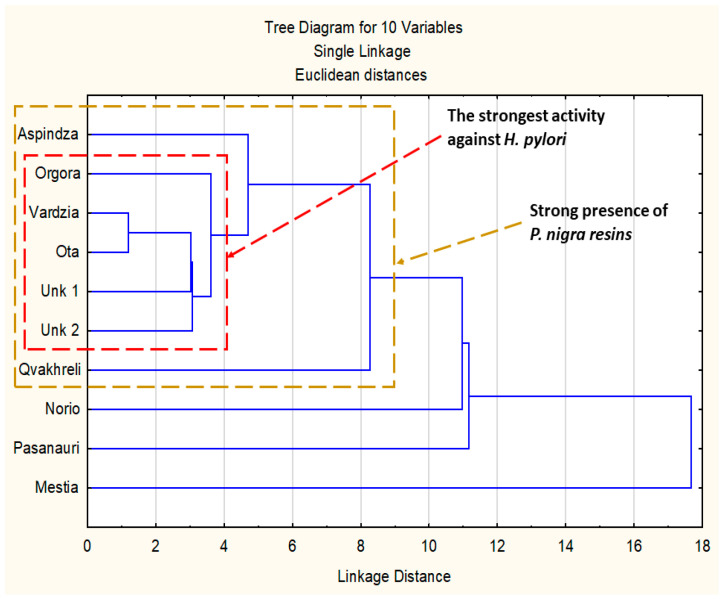
Hierarchical fuzzy clustering analysis presented as a dendrogram.

**Table 1 molecules-28-01374-t001:** Antimicrobial properties of Georgian propolis extracts against the reference *H. pylori* ATCC 43504 and clinical *H. pylori* strains.

	*H. pylori* ATCC 43504		*H. pylori* Clinical Strains
Propolis Sample	MIC	MBC	MIC_50_/_90_
Norio	125.0	125.0	250/250
Pasanauri	62.5	62.5	62.5/125
Qvakhreli	62.5	62.5	62.5/62.5
Ota	31.3	31.3	31.3/31.3
Ogora	31.3	31.3	15.6/31.3
Aspindza	62.5	62.5	31.3/62.5
Vardzia	31.3	31.3	31.3/31.3
Mestia	125.0	125.0	125/250
Unknown 1 (Kakhetia)	31.3	31.3	31.3/31.3
Unknown 2 (Kakhetia)	31.3	31.3	31.3/31.3

MIC (minimal inhibitory concertation) and MBC (minimal bactericidal concentration), was described in µg/mL.

**Table 2 molecules-28-01374-t002:** Antibacterial activity (MIC) and inhibition of *H. pylori* urease (IC_50_) by tested Georgian propolis extracts.

Propolis Sample	IC_50_ (µg/mL)	% Inhibition of Control in 1 mg/mL Concentration
Ota	549.9	64.2%
Vardzia	1484.8	66.6%
Orogora	864.7	64.7%
Norio	212.82	66.0%
Pasanauri	962.18	42.5%
Qvakhreli	141.57	74.7%
Aspindza	317.15	78.3%
Mestia	1594.67	51.9%
Unknown 1 (Kakhetia)	3859.23	70.1%
Unknown 2 (Kakhetia)	4.01	71.0%
Thiourea	92.7	100%

**Table 3 molecules-28-01374-t003:** Main components of Georgian propolis extracts ^†^.

	RT MS	UV max [nm]	[M − H^+^]^−^	ASP	NOR	PAS	MES	ORG	VAR	OTA	QVA	UNK1	UNK2
Vanillin isomer ^b,c^	9.34	**310,280**	151.0393	0.03	0.58	1.60	2.87	0.05	0.06	0.14	—	0.02	0.02
Caffeic acid ^a,b,c^	11.56	**323**	179.0346	2.03	4.31	2.53	1.19	3.11	3.28	2.85	4.30	3.47	3.78
*p*-Coumaric acid ^a,b,c^	14.45	**310**	163.0401	1.37	5.41	8.36	10.44	2.32	2.25	2.23	2.72	2.23	1.99
Ferulic acid ^a,b,c^	15.24	**325**	193.0504	0.61	1.08	4.59	9.77	0.45	0.38	0.38	0.46	0.68	0.64
Isoferulic acid ^a,b,c^	15.75	**324**	193.0503	1.33	5.86	2.20	0.07	3.13	3.14	3.15	3.97	2.86	2.49
Caffeic acid ethyl ester ^b,c^	19.52	**321**	207.0662	0.30	3.61	2.55	—	1.73	1.51	1.58	2.25	1.40	1.16
^iw^ Cinnamic acid ^a,b,c^	21.36	**280**	—	0.24	1.35	0.82	0.77	0.28	0.44	0.46	0.21	0.37	0.30
Unidentified	23.24	**308**	—	0.30	2.09	1.41	—	1.18	0.89	0.96	1.61	0.73	0.63
Pinobanksin 5-methylether ^b,c^	23.54	**287**	285.0777	2.86	0.89	0.99	—	1.93	2.71	2.64	0.96	2.70	2.88
Unidentified	26.00	**265**	—	—	—	0.84	1.40	—	0.07	0.06	0.04	—	—
Pinobanksin ^a,b,c^	27.45	**292**	271.0615	5.24	2.70	1.70	—	3.60	4.41	4.17	2.47	3.80	4.52
Apigenin ^a,b,c^	30.66	**338**, 263	269.0457	1.75	0.76	1.11	1.02	1.22	1.14	1.15	1.22	1.18	1.39
Unidentified	31.92	**310**	—	1.00	—	0.34	—	0.57	—	0.57	0.43	0.78	0.81
1,3-di-*p*-Coumaroylglycerol ^b,c^	33.98	**312**	383.1143	—	—	0.52	1.89	—	—	—	—	—	—
(R/S) 1-*p*-Coumaroyl-3-feruloylglycerol ^b,c^	34.48	**316**	413.1241	—	—	0.40	1.11	—	—	—	—	—	—
Caffeic acid 2-methyl-2-butenyl ester ^b,c^	39.50	**325**	247.0979	2.41	3.97	0.70	—	5.62	3.21	3.27	9.69	4.39	3.79
Caffeic acid 3-methyl-2-butenyl ester (Basic prenyl ester) ^b,c^	40.91	**325**	247.0979	3.63	4.97	1.18	—	8.05	4.21	4.33	13.60	5.67	5.15
Caffeic acid 3-methyl-3-butenyl ester ^b,c^	41.42	**325**	247.0977	0.28	0.52	—	—	0.77	0.43	0.39	1.19	0.48	0.42
(R/S) 2-Acetyl-1-caffeoyl-3-*p*-coumaroylglycerol ^b,c^	41.91	**315**	441.1197	—	0.26	0.47	1.28	—	—	—	—	—	—
Chrysin ^a,b,c^	42.38	312sh, **268**	253.0505	14.69	8.02	6.81	—	12.38	11.98	12.07	11.34	12.39	12.64
Caffeic acid benzyl ester ^b,c^	42.69	**326**	269.0818	2.31	4.44	3.10	—	2.66	3.06	3.08	2.56	2.73	2.67
(R/S) 2-Acetyl-1-caffeoyl-3-feruloylglycerol ^b,c^	42.71	**325**	471.1297	—	—	—	1.71	—	—	—	—	—	—
* Sakuranetin isomer ^c^	43.29	**287**	285.0769	—	—	—	2.07	—	—	—	—	—	—
Pinocembrin ^b,c^	43.41	**290**	255.0666	9.72	6.59	3.84	—	9.84	11.84	11.19	9.09	9.32	9.65
Sakuranetin ^b,c^	44.69	**290**	285.0773	2.15	0.30	0.55	1.62	1.66	1.92	2.02	1.22	1.84	2.00
Galangin ^a,b,c^	45.17	360, **266**	269.0454	7.13	2.50	1.63	—	5.32	7.30	7.01	3.38	6.37	6.62
Acacetin ^a,b,c^	45.78	335, **269**	283.0614	0.70	0.28	0.97	2.27	0.56	0.49	0.53	0.57	0.62	0.59
Ermanin isomer ^b,c^	46.13	333, **275**	313.0721	—	—	3.52	8.09	—	—	—	—	0.17	0.22
Caffeic acid phenethyl ester (CAPE) ^b,c^	47.21	326	283.0981	2.22	2.53	1.57	—	2.35	1.85	1.94	2.99	2.16	2.17
Pinobanksin 3-*O*-acetate ^b,c^	47.69	**295**	313.0725	13.35	7.09	5.73	—	10.52	11.20	11.50	7.94	12.00	12.31
Methoxychrysin ^b,c^	48.06	310sh, **266**	283.0614	2.03	0.15	—	—	0.77	0.59	0.32	0.72	0.35	0.70
Quercetin-dimethyl ether ^b,c^	48.19	#**370**	329.0667	—	—	0.75	1.59	—	—	—	—	—	—
2-Acetyl-1,3-di-*p*-coumaroylglycerol ^b,c^	50.93	**312**	425.1242	0.50	1.92	3.33	9.39	—	—	—	—	—	0.70
(R/S) 2-Acetyl-3-*p*-coumaroyl-1-feruloylglycerol ^b,c^	51.87	**316**	455.1336	—	0.99	3.23	7.83	—	—	—	—	—	—
*p*-Coumaric acid 3-methyl-2-butenyl or 2-methyl-2-butenyl ester ^b,c^	52.18	**313**	231.1027	1.23	1.12	1.28	—	0.10	0.41	0.40	0.61	0.57	1.32
2-Acetyl-1,3-di-feruloylglycerol ^b,c^	52.49	**324**	485.1456	—	0.64	2.14	3.56	—	—	—	—	—	—
*p*-Coumaric acid benzyl ester ^b,c^	53.88	**316**	253.0869	—	1.80	5.15	5.40	1.54	1.37	1.35	1.27	1.40	—
(R/S) 1-Acetyl-2,3-di-feruloylglycerol ^b,c^	53.9	**324**	485.1455	1.30	0.80	2.41	2.6	1.80	2.11	2.08	1.04	2.05	—
Caffeic acid cinnamyl ester ^b,c^	56.10	**323**	295.0982	0.63	3.39	1.96	—	0.96	1.52	1.60	0.54	1.36	1.31
Pinobanksin-3-*O*-propanoate ^b,c^	58.20	**294**	327.0878	1.35	0.24	0.82	—	0.61	0.50	0.52	0.23	0.83	0.90
^iw^ Tectochrysin	63.00	313, **268**	—	2.12	0.97	0.70	—	1.48	1.11	1.25	1.21	1.34	1.48
^iw^ Pinostrobin	63.48	**288**	—	1.06	1.01	1.45	—	1.11	1.61	1.60	0.68	1.03	1.01
*p*-Coumaric acid cinnamyl ester ^b,c^	64.11	**313**	279.1029	0.18	2.30	2.30	—	0.43	0.57	0.77	0.13	0.44	0.56
Unidentified	64.6	**323**	—	0.80	3.43	2.38	—	1.10	1.53	1.76	0.63	1.44	1.51
Pinobanksin 3-*O*-butanoate or isobutanoate ^b,c^	64.92	**293**	341.1037	1.22	0.26	0.46	—	0.73	0.77	0.84	0.30	0.82	0.95
Unidentified	67.34	**279**	—	0.15	1.47	2.02	7.70	0.06	—	—	—	—	—
*p*-Metoxycinnamic acid cinnamyl ester ^b,c^	69.35	**282**	293.2125	2.53	0.63	1.82	—	1.90	1.84	1.92	0.74	1.73	1.89

^†^ Amount of single component, calculated as percentage of whole UV chromatogram area (280 nm); UV max [nm]—maximum of UV absorption, higher maxima bolded; —component did not produce ion or did not have UV spectrum (or too low concertation). ^a^ Component identified by comparison with standard; ^b^ component identified by comparison with literature; ^c^ component identified by prediction of mass fragment and UV spectrum; * component tentatively identified; ^iw^ component does not produces or produce low/trace amount of ions in negative mode ASP—Aspindza; NOR—Norio; PAS—Pasanauri; MES—Mestia; ORG—Orgora; VAR—Vardzia; OTA—Ota; QVA—Qvakhreli; UNK1—unknown region 1 in Kakhetia; UNK2—unknown region 2 in Kakhetia.

## Data Availability

Not applicable.

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
