# Peer review of "Correlation between Chemical Profile of Georgian Propolis Extracts and Their Activity against Helicobacter pylori"

_molecules, 2023, doi:10.3390/molecules28031374_

Round 1

Reviewer 1 Report

The manuscript molecules-2168376 "Correlation between chemical profile of Georgian propolis extracts and their activity against Helicobacter pylori" provides an up-to-date and important information on the effects of bee products (propolis) against H. pylori. The main goals of the study is to evaluate the anti-H. pylori activity of tested propolis samples obtained from different location of Georgia; the assessment of the relation between the polyphenolic profile of propolis, plant origin and antimicrobial activity against H. pylori and to evaluate the inhibition activity of selected propolis samples toward urease enzyme.

This data is relevant for a broad public since the incidence of diseases caused by H. pylori has increased significantly in recent years and natural treatments are preferable to synthetic ones. It is perfectly clear to anyone familiar with this type of research that it had to be a great effort for authors to plan and conduct such a study, and I have a great appreciation for this. In my opinion, the writing manner is very effective in uptake of prominent and key points of each research work, the way presentation of the manuscript is very well, however grammatical and technical error issue arises and it may affect the manuscript quality. Thus, the authors need to pay much attention to eliminate this kind of errors to provide a good quality to the paper for publication. The background provides sufficient literature review, the methodology is sound, results are explanatory and well-discussed. Overall, a good read. I fully support the publication of this paper in Molecules after a minor revision.

Author Response

Reviewer 1

We would like to thank the Reviewer for the suggestions for revision, which undoubtedly improved our paper.

We have revised the manuscript in accordance with the comments of the reviewers, and we enclose a revised manuscript with introduced changes highlighted in yellow. We introduced minor modifications in the Introduction, Methods and the Results and Discussion sections, based on all reviewers’ comments. A few references ware also added.

Please find below detailed responses to each of your comments indicating what has been revised.

The manuscript molecules-2168376 "Correlation between chemical profile of Georgian propolis extracts and their activity against Helicobacter pylori" provides an up-to-date and important information on the effects of bee products (propolis) against H. pylori. The main goals of the study is to evaluate the anti-H. pylori activity of tested propolis samples obtained from different location of Georgia; the assessment of the relation between the polyphenolic profile of propolis, plant origin, and antimicrobial activity against H. pylori and to evaluate the inhibition activity of selected propolis samples toward urease enzyme.

This data is relevant for a broad public since the incidence of diseases caused by H. pylori has increased significantly in recent years and natural treatments are preferable to synthetic ones. It is perfectly clear to anyone familiar with this type of research that it had to be a great effort for authors to plan and conduct such a study, and I have a great appreciation for this. In my opinion, the writing manner is very effective in uptake of prominent and key points of each research work, the way presentation of the manuscript is very well, however grammatical and technical error issue arises and it may affect the manuscript quality. Thus, the authors need to pay much attention to eliminating these kinds of errors to provide good quality to the paper for publication. The background provides a sufficient literature review, the methodology is sound, results are explanatory and well-discussed. Overall, a good read. I fully support the publication of this paper in Molecules after a minor revision.

Answer: Thank you very much for this opinion. The manuscript was extensively corrected by English native speaker.

Reviewer 2 Report

The authors studied the effect of ethanol extracts of Georgian propolises on Helicobacter pylori. They identified samples with high activity and correlated these samples compositions with black poplar origin. The results are clearly presented. The discussion, mixed with the results in the manuscript, is interesting, providing an overview of the all the studies of propolises effects on Helicobacter pylori. The authors could discuss a little more the effects of propolis on peptidoglycan and compare effects on Gram positive and Gram negative bacteria.

However the manuscript needs extensive English editing for grammar, style and misspelling because in the present form it is difficult to read. The authors should carefully correct the manuscript.

Here are some examples of things which should be corrected:

For instance, line 30 in the abstract, "The" should be replace by "we"? line 43-44 sentence has to be reformulated, there are to occurrences of "in in", line 102 "showed" instead of "shown", "stains" line 127, "assessment" instead of "assessed line 170, "my" line 129 etc...

Line 112-113: the sentence is misleading and contradictory to lines 133-135. The authors should add ..." 'Georgian' propolis originated from Georgia".

Figure 3: labels are very difficult to read

Figure 4 is labelled Figure 3

Line 403: FC is for TF? It is not clear

Author Response

Reviewer 2

We would like to thank the Reviewer for the suggestions for revision, which undoubtedly improved our paper.

We have revised the manuscript in accordance with the comments of the reviewers, and we enclose a revised manuscript with introduced changes highlighted in yellow. We introduced minor modifications in the Introduction, Methods and the Results and Discussion sections, based on all reviewers’ comments. A few references were also added.

Please find below-detailed responses to each of your comments indicating what has been revised

The authors studied the effect of ethanol extracts of Georgian propolis on Helicobacter pylori. They identified samples with high activity and correlated these sample's compositions with black poplar origin. The results are clearly presented. The discussion, mixed with the results in the manuscript, is interesting, providing an overview of all the studies of propolis effects on Helicobacter pylori. The authors could discuss a little more the effects of propolis on peptidoglycan and compare effects on Gram-positive and Gram-negative bacteria.

However, the manuscript needs extensive English editing for grammar, style, and misspelling because, in the present form, it is difficult to read. The authors should carefully correct the manuscript.

Answer: Thank you very much for this opinion. The manuscript was corrected by an English native speaker.

Here are some examples of things that should be corrected:

For instance, in line 30 in the abstract, "The" should be replaced by "we"? line 43-44 sentence has to be reformulated, there are two occurrences of "in in", line 102 "showed" instead of "shown", "stains" line 127, "assessment" instead of "assessed line 170, "my" line 129, etc...

Answer: The corrections have been made.

Line 112-113: the sentence is misleading and contradictory to lines 133-135. The authors should add ..." 'Georgian' propolis originated from Georgia".

Answer: The corrections have been made.

Figure 3: labels are very difficult to read

Answer: The corrections have been made.

Figure 4 is labeled Figure 3

Answer: The correction has been made.

Line 403: FC is for TF? It is not clear

Answer: The correction has been made.

Reviewer 3 Report

Check the manuscript carefully there are several language and syntax mistakes in the manuscript

Conclussion is written superficially, please reconsider it, adnd describe the results with more details

One cannot understand nothing from Figure 2, try to enlarge it to make text more visible

In introduction section describe propolis properties with more details. Helicobacter pylori is described in detail but only one phrase about propolis is mentioned

Which are the particularities of this type of propolis? What are the differences between this type and other types of propolis?

All the medical uses of propolis are potential uses, nothing is demonstrated through clinical trials. The author's should take into consideration this fact.

Author Response

Reviewer 3

We would like to thank the Reviewer for the suggestions of revision, which undoubtedly improved our paper.

We have revised the manuscript in accordance with the comments of the reviewers, and we enclose a revised manuscript with introduced changes highlighted in yellow. We introduced minor modifications in the Introduction, Methods and the Results and Discussion sections, based on all reviewers’ comments. A few references were also added

Please find below-detailed responses to each of your comments indicating what has been revised.

Check the manuscript carefully there are several language and syntax mistakes in the manuscript

Answer: Thank you very much for this opinion. The manuscript was corrected by an English native speaker.

The conclusion is written superficially, please reconsider it, and describe the results in more details.

Answer: Thank you for this suggestion. It is corrected.

One cannot understand nothing from Figure 2, try to enlarge it to make text more visible

Answer: The corrections have been made.

In introduction section describe propolis properties with more details. Helicobacter pylori is described in detail but only one phrase about propolis is mentioned.

Answer: Thank you for this suggestion. The information about propolis antioxidant and anti-inflammatory activity were added.

Which are the particularities of this type of propolis? What are the differences between this type and other types of propolis?

Answer: Thank you for this suggestion. Tested Georgian propolises exhibited uHPLC-DAD-MS/MS profile characteristics for strong black poplar origin, presence of strong P. nigra markers peaks, and lack of another specific marker. Observed black poplars markers included flavonoids (chrysin, pinocembrin, galangin and pinobanskin and their esters, especially 3-O-pinobanksin acetate) as well as phenolic acids monoesters (mainly ester of caffeic acid such as 3-methyl-2-butenyl, 2-methyl-2-butenyl and phenethyl). Apart from same presence of common components, important is also their concentration – black poplars are known for the relatively high or high presence of free phenolic acid, while in aspen they are minor components and are trace or absent in birches.

All the medical uses of propolis are potential uses, nothing is demonstrated through clinical trials. The author's should take into consideration this fact.

Answer: Thank you for this suggestion. The study we carried out was the very first insight into the anti-H. pylori activity of Georgian propolis. We are planning to extend the range of experiments including initial pre-clinical phase trials.

Reviewer 4 Report

The manuscript entitled "Correlation between chemical profile of Georgian propolis extracts and their activity against Helicobacter pylori" is a scientifically driven work. Although the work is of great significance, there are many modifications and clarity is needed in several parts. Please find my comments below

1. Abstract line 21-23 must be restructured

2. Line 26-27 MIC and MBC same value?

3. Chemical composition of propolis can be included in abstract

4. The introduction needs to emphasize the impact of H. pylori on gastric damage and carcinogenesis with its Molecular Mechanisms involved.

5. There are several studies on the potential of propolis against H pylori. There is no mention about it.

https://doi.org/10.1016/j.bjp.2019.03.002, https://doi.org/10.1007/s12275-020-0277-z, https://doi.org/10.3390/nu14214644

6. Compared to the above mentioned studies, what is the novelty of the present article? Authors must emphasize the same in the introduction

7. Why the authors represented MIC / MBC values without standard deviation or SE in table 1

8. Same way IC50 values also lack SD/SE values

9. In methods, the UPLC conditions can be mentioned briefly

10. I suggest to include the disc Diffusion assay to determine the zone of inhibition

11. The result of DPPH or FRAP was not represented in the manuacript

Author Response

Reviewer 4

We would like to thank the Reviewer for the suggestions of revision, which undoubtedly improved our paper.

We have revised the manuscript in accordance with the comments of the reviewers, and we enclose a revised manuscript with introduced changes highlighted in yellow. We introduced minor modifications in the Introduction, Methods and the Results and Discussion sections, based on all reviewers’ comments. A few references were also added.

Please find below detailed responses to each of your comments indicating what has been revised.

The manuscript entitled "Correlation between the chemical profile of Georgian propolis extracts and their activity against Helicobacter pylori" is a scientifically driven work. Although the work is of great significance, there are many modifications, and clarity is needed in several parts. Please find my comments below

  1. Abstract line 21-23 must be restructured

Answer: Thank you for this suggestion. The corrections have been made.

  1. Line 26-27 MIC and MBC same value?

Answer: Thank you for this suggestion. Yes, they are. It means that MBC/MIC index equals 1 for these propolises and is revealing their bactericidal activity against H. pylori.

  1. The chemical composition of propolis can be included in abstract

Answer: Thank you for this suggestion. Data concerning the chemical composition of tested propolis extracts were included in the abstract.

  1. The introduction needs to emphasize the impact of H. pylori on gastric damage and carcinogenesis with its Molecular Mechanisms involved.

Answer: Thank you for this suggestion. The severity and progression of gastric cancer depend on the presence of specific H. pylori virulence factors as well as H. pylori infection influences cellular components that are associated with epithelial–mesenchymal transition progression. [Baj, J.; Korona-Głowniak, I.; Forma, A.; Maani, A.; Sitarz, E.; Rahnama-Hezavah, M.; Radzikowska, E.; Portincasa, P. Mechanisms of the Epithelial–Mesenchymal Transition and Tumor Microenvironment in Helicobacter pylori-Induced Gastric Cancer. Cells 2020, 9, 1055. https://doi.org/10.3390/cells9041055]. This was added to the manuscript.

  1. There are several studies on the potential of propolis against H pylori. There is no mention about it.

https://doi.org/10.1016/j.bjp.2019.03.002,

https://doi.org/10.1007/s12275-020-0277-z,

https://doi.org/10.3390/nu14214644

Answer: Thank you for this suggestion. The first recommended paper is already mentioned in our paper; references position number 20. The other two articles related to the anti-inflammatory properties of Korean propolis, which is an interesting topic but only indirectly related to the subject of the reviewed work. However, it is a valuable suggestion for further research directions. Therefore, these two articles are cited in the introduction.

  1. Compared to the above-mentioned studies, what is the novelty of the present article? Authors must emphasize the same in the introduction.

Answer: Thank you for this suggestion. The presented article is the first report of activity against H. pylori of extracts obtained from propolis samples collected in different regions of Georgia. The tested extracts are characterized by a low MIC value and a highly promising urease inhibition ability.

  1. Why the authors represented MIC / MBC values without standard deviation or SE in table 1

Answer: Thank you for this suggestion. It is rather unusual to express the MIC or MBC values with standard deviations. Indeed, we tested the extracts in triplicate, and in all but a few cases we observed the same MIC values. According to European Committee on Antimicrobial Susceptibility Testing (EUCAST) recommendation, representative values (the modes) were presented.

  1. Same way IC50 values also lack SD/SE values

Answer: Thank you for this suggestion.

The following equation was employed to calculate the enzymatic reaction value:

I (%) = [1 – (average absorbance of propolis extract in the presence of enzyme - average absorbance of the solvent with extract but without enzyme/ positive control in the solvent in the presence of enzyme - average absorbance of the solvent without enzyme)] * 100

Data of the absorbance were expressed as mean ± standard error (SD) and the average results were taken from at least three measurements to IC50 calculation in online calculator - AAT Bioquest.

  1. In methods, the UPLC conditions can be mentioned briefly

Answer: Thank you for this suggestion. A description of UHPLC-DAD-MS/MS was added.

  1. I suggest including the disc Diffusion assay to determine the zone of inhibition

Answer: Thank you for this suggestion. The antimicrobial activity testing was performed according to (EUCAST) recommendations. According to EUCAST, the disc diffusion method for fastidious bacteria such as H. pylori is not recommended – therefore we did not use it.

  1. The result of DPPH or FRAP was not represented in the manuscript

Answer: Thank you for this suggestion. The results of antioxidant assay (DPPH as well FRAP) are presented in Supplementary Materials.

Round 2

Reviewer 4 Report

The manuscript is improved in quality compared to precious version. Authors need to take care about the punctuation errors and typographic mistakes. Also, I recommend to conduct more Molecular analysis in the future studies to make the article more informative.